# Secreted Amyloid Precursor Protein Alpha (sAPPα Regulates the Cellular Proteome and Secretome of Mouse Primary Astrocytes

**DOI:** 10.3390/ijms24087165

**Published:** 2023-04-12

**Authors:** Katie Peppercorn, Torsten Kleffmann, Stephanie M. Hughes, Warren P. Tate

**Affiliations:** 1Department of Biochemistry, School of Biomedical Sciences, Division of Health Sciences, University of Otago, Dunedin 9016, New Zealand; 2Brain Health Research Centre, University of Otago, Dunedin 9016, New Zealand; 3Research Infrastructure Centre, Division of Health Sciences, University of Otago, Dunedin 9016, New Zealand; 4Genetics Otago, University of Otago, Dunedin 9016, New Zealand

**Keywords:** APP, sAPPα, Alzheimer’s disease, astrocytes, proteome, secretome

## Abstract

Secreted amyloid precursor protein alpha (sAPPα), processed from a parent mammalian brain protein, amyloid precursor protein, can modulate learning and memory. Recently it has been shown to modulate the transcriptome and proteome of human neurons, including proteins with neurological functions. Here, we analysed whether the acute administration of sAPPα facilitated changes in the proteome and secretome of mouse primary astrocytes in culture. Astrocytes contribute to the neuronal processes of neurogenesis, synaptogenesis and synaptic plasticity. Cortical mouse astrocytes in culture were exposed to 1 nM sAPPα, and changes in both the whole-cell proteome (2 h) and the secretome (6 h) were identified with Sequential Window Acquisition of All Theoretical Fragment Ion Spectra–Mass Spectrometry (SWATH-MS). Differentially regulated proteins were identified in both the cellular proteome and secretome that are involved with neurologically related functions of the normal physiology of the brain and central nervous system. Groups of proteins have a relationship to APP and have roles in the modulation of cell morphology, vesicle dynamics and the myelin sheath. Some are related to pathways containing proteins whose genes have been previously implicated in Alzheimer’s disease (AD). The secretome is also enriched in proteins related to Insulin Growth Factor 2 (IGF2) signaling and the extracellular matrix (ECM). There is the promise that a more specific investigation of these proteins will help to understand the mechanisms of how sAPPα signaling affects memory formation.

## 1. Introduction

Astrocytes are a glial cell type essential to the proper functioning of the brain, contributing to neurogenesis, synaptogenesis [1], synaptic plasticity [2] and memory encoding [3]. Astrocytes can associate with many neurons and thereby thousands to millions of synapses [4,5,6], where they communicate with neurons bidirectionally [7]. Astrocytes can modulate neuronal function at the network level, where they are positioned to respond to neural network activity and impose a significant impact on that activity under both physiological and pathological conditions [8].

Astrocytes are morphologically plastic and able to transform shape quickly [9]. One of four morphologically distinct astrocytes, protoplasmic astrocytes [10], can enwrap fine processes called perisynaptic astrocytic processes (PAPs) around individual synapses, isolating them and preventing spillover of neurotransmitters to neighbouring synapses while providing essential biochemical support, enabling the modulation of synaptic transmission [11], cognition and information processing [2]. The PAP membrane contains glutamate transporters, which remove glutamate from the synaptic cleft to terminate transmission and the proteins required for releasing fast transmitters (e.g., glutamate, ATP, D-serine and GABA) [12].

Dysfunctional astrocytes have been linked to several neurological diseases, including Alzheimer’s disease (AD), Huntington’s disease, amyotrophic lateral sclerosis, epilepsy and stroke [13]. A reduction in astrocyte volume, surface area and morphological complexity has been observed in AD mouse models [14,15], and in human patients, this reduction correlates with increased dementia severity [16].

Astrocytes’ contribution to neuronal synaptogenesis [1] is achieved through the secretion of molecules, including proteins, into the extracellular milieu [17]. This comprises the cellular secretome, which is defined as the fraction of proteins, peptides and small molecules secreted from the cell. It includes growth factors, matricellular proteins, neurotransmitters, cytokines and chemokines and has been investigated in mice [18,19,20], rats [21] and humans [17]. Extracellular vesicles (EVs) derived from astrocytes are a key component of the secretome and contain molecules destined for specific target cells [22]. The contents of EVs are altered due to neurodegenerative disease [23].

Secreted Amyloid Precursor Protein Alpha (sAPPα) is a multifunctional protein with a predominant neural isoform, sAPPα^695^, also referred to as sAβPPα or APPsα. It is released into the extracellular space after the cleavage of the integral membrane parent protein Amyloid Precursor Protein (APP) by alpha secretase [24]. APP is a cell adhesion molecule [25] associated with synaptic plasticity [26], synapse formation and stability [27], and it is also essential for the normal development of the CNS [28]. sAPPα can restore the ability to form memories [29,30,31], and it prevents the development of memory defects in AD mouse models [32,33]. The electrophysiological model of learning, long-term potentiation (LTP), is modulated by sAPPα [26,30,31,32,34,35]. sAPPα promotes neurogenesis [36,37], neural cell proliferation [38] and can modulate dendritic spine density [39] and neurite outgrowth [40,41,42,43]. sAPPα also confers neuroprotection against hypoglycemia [44], amyloid beta (Aβ) toxicity [45] and glutamate toxicity [45].

Multiple receptors for sAPPα have been identified in neurons that are also expressed in astrocytes, including gamma-aminobutyric acid type B receptor subunit 1a (GABABR1a) [46,47], nicotinic a7-nACh receptors (a7-nAChRs) [39,48], sortilin [49,50], sorting-protein-related receptor A (SORLA) [51,52] and APP itself [53]. Nevertheless, no information concerning how sAPPα affects astrocytes is currently available.

A number of genes and proteins have been identified in *rodent* neurons that are regulated by sAPPα [54,55,56], and we recently determined how sAPPα affected the transcriptome and proteome of *human* glutamatergic neurons in culture [57]. Differentially regulated proteins include those encoded by AD risk genes, APP-processing-related proteins, proteins involved in synaptogenesis, neurotransmitter receptors, synaptic vesicle proteins and cytoskeletal proteins.

We hypothesised that sAPPα would regulate the neurological functions of the astrocyte via changes to its proteome. With the support that the astrocyte provides to the functions of the neuron, we further hypothesised that this regulation would include effects on the secretome of the astrocyte. Therefore, in the current study, we aimed to answer (i) whether sAPPα affects the molecular biology of the astrocyte, (ii) whether it changes the secretome as a consequence of changing the cellular proteome and (iii) whether any observed changes are related to neurological functions. We have shown that acute short-term exposure of mouse primary astrocytes to exogenous sAPPα^695^ resulted in significant changes in the proteins that are important for neurological function, both in the cellular proteome and, subsequently, in the secretome. Groups of proteins that show changes in both the proteome and secretome have a relationship to APP and have roles in cell morphology, vesicle dynamics and the myelin sheath. Some are in the Alzheimer’s disease KEGG pathway (mmu05010). This pathway includes the ‘risk genes’, for example, PSEN2, which predisposes to Alzheimer’s disease (AD) and that was identified in the dataset.

## 2. Results

### 2.1. Immunological Characterisation of the Cells

Immunocytochemistry confirmed that the primary cell cultures were highly enriched in astrocytes (Figure 1A–C) as the cells were positive for the astrocyte marker proteins of GFAP, s100b or both when tested together. Astrocytes with different morphological forms were present in the culture (Appendix A), as might be expected in primary cells from mouse brains. Low-level microglial contamination was observed, with 8–10% of the nuclei from the four biological replicates staining positive for the microglial marker of CD68 (Figure 1D, white arrows). While this may have some influence on the proteome data, it was regarded as an acceptable level of contamination that would not undermine the validity of the analysis of astrocyte function. sAPPα can induce activation pathways in microglia [58]. In all images, the nuclei were stained with DAPI (blue).

### 2.2. Astrocyte Cellular Proteome Analysis after 2 h Exposure to Exogenous sAPPα

The proteome of the mouse primary astrocytes was analysed using Sequential Window Acquisition of all Theoretical Fragment Ion Spectra–Mass Spectrometry (SWATH-MS) following a 2 h incubation with sAPPα to identify proteins that were differentially regulated immediately in response to treatment.

#### 2.2.1. The Spectral Library

The DDA (Data Dependent Acquisition) of the pre-fractionated pooled sample identified 6551 proteins and 154,232 peptides at a confidence interval of ≥99 and an FDR of q ≤ 0.01, which were integrated into the spectral library. After aligning the DIA data to the spectral library, 4256 proteins and 14,309 peptides were used for quantification based on an FDR for a peak matching of q ≤ 0.01 in at least one sample.

#### 2.2.2. Changes in Astrocyte Cellular Proteome

At the protein level, using the average of all the peptide intensities of each protein, 64 proteins were identified using the spectral library as being differentially regulated, with 24 proteins down-regulated (log2(FC) ≤ −0.58, −log10(*p*-value) ≥ 1.3) and 40 up-regulated (log2(FC) ≥ 0.58, −log10(*p*-value) ≥ 1.3), as delineated by the orange lines in the volcano plots, with significantly differentially regulated (SDR) proteins shown as orange points (Figure 2A). All of the proteins that were significantly differentially regulated are listed in Appendix A.

At the peptide level, where individual peptide intensities were allowed for quantification, an expanded set of 341 peptides matched to 271 proteins showed a significant difference in relative abundance (log2(FC) ≤ −0.58 or ≥ 0.58, −log10(*p*-value) ≥ 1.3). It should be noted that this still required the data in each case to be closely concordant with the three biological replicate samples and from three technical replicates from each of these samples. From this analysis, now 115 proteins were down-regulated, and 156 were up-regulated (Figure 2B and Appendix A).

A STRING functional network analysis was performed on the differentially expressed proteins identified from more than one peptide (Figure 2A and Appendix A). APP (from which the protein sAPPα is derived) was added as an additional node. There were three main clusters in the network, with the functions of (i) chromatin organization (GO:0006325; strength 1.29; FDR 0.0092), (ii) RAB protein signal transduction (GO:0032482; strength 2.43; FDR 5.98 × 10^−6^) and (iii) membrane protein ectodomain proteolysis (GO:0006509; strength 2.72; FDR 0.0486). The latter cluster includes APP as the central node with a first-level interaction (edges) with three SDR proteins, PSEN2, KLC2 and ADAM9. PSEN2 and ADAM9 have protease activities and are involved in regulatory proteolytic processes.

#### 2.2.3. Differentially Regulated Proteins of the Proteome Identified by an Individual Peptide

STRING network analysis and pathway enrichment analysis of the 270 proteins differentially regulated identified the networks and biological pathways in which these proteins were involved (Appendix A). Gene Ontology (GO) categories, Reactome and KEGG pathways enrichment analysis revealed that sAPPα-stimulated astrocytes up-regulated the expression of the proteins important for neurological function (Table 1). Of these proteins, 48 are related to the GO cellular component category ‘synapse; PHB2, NDRG2, MAP2K1, APP, PSEN2, KPNA2, YWHAH, PPP2CA, DBN1, AP3B1, NEFM, GSK3B, BIN1, FARP1, SEPT2, DARS, LAMC1, WASL, GPC4, XRN1, ARHGAP39, CDC42, RPL18A, ITGB5, RPL26, RPS7, SEPT11, ENAH, ERC1, RPL17, RPL29, RPLP2, NPTN, TUFM, ELAVL1, EPB4.1, FUS, SNX27, CLTA, RPS19, DMD, ITSN1, ADD1, HNRNPA2B1, GLS, SRI, SH3GLB1 and ERBB2IP (See Tables below for full names). A comparison of our differentially regulated proteins with the SynGO resources [59] revealed 47 proteins with synapse functions.

The added node APP was also part of the proteins associated with the GO-term ‘synapse’ and showed 15 first-level interactions (Figure 3), including the three proteins already identified in the protein-level analysis (ADAM9, KLC2, PSEN2, CAV1, SNX27, FUS, GSK3, DBN1, STUB1, ERC1, SOD1, SNX17, BIN1, GSK3B, NFIB). The evidence for the direct associations is shown in Table 2, with the scores indicating the strength of the likelihood of an interaction. Pathway analysis of this subset of regulated proteins showed an enrichment of protein nodes being assigned to GO terms, such as the regulation of proteolysis (GO:0030162; strength 1.11; FDR 0.00036) and the positive regulation of protein catabolic processes (GO:0045732; strength 1.55; FDR 7.33 × 10^−5^), therefore, adding further significance to the associated functions of SDR proteins identified in the protein-level analysis.

Network clustering of the peptide-level data applying the Markov Clustering Algorithm (MCL) [60] with an inflation parameter of 1.5 revealed four clusters of functionally related proteins (Figure 4): Cluster 1 with a group of neurologically related functions; Cluster 2, mitochondrial functions and energy production; Cluster 3, a collection of ribosome- and RNA-related proteins related to ribosomal RNA; and Cluster 4 proteins, associated with the microtubule cytoskeleton and the cell cycle.

Significantly, 66 of the 156 proteins that were up-regulated and 42 of the 115 proteins that were down-regulated were functionally associated with either APP, neurological functions or AD (Table 3).

In Table 3, we have listed all of the proteins that were differentially up- or down-regulated in the proteome that were identified under a specific GO term related to neurological function (synapse, neuron projection, myelin sheath and cytoskeleton) or to APP (sAPPα), and those within a pathway linked to AD. Most of the differentially regulated proteins scored in more than one category, and approximately 40% of all of the regulated proteins were identified in these categories.

Ribosomal proteins, for example, L26 and S7, which were found in our data set (Table 3), can have extra-ribosomal functions in signaling pathways [61]. L26 has been found to influence events, such as the activation of p53, and S7 influences PI3K/AKT and MAPK signaling. This may occur in the astrocyte itself. Therefore, while these proteins from within the neuron may influence a signaling pathway that is directly linked to synapse formation or neuronal projections in the neuron, as identified in the literature, they could have other neurologically important functions in the astrocyte. Although still poorly understood, there are translationally active microzones in neurons at the synapses and dendrites (neuronal projections). Specific ribosomal proteins in their ribosomal complexes may play a unique role in these.

### 2.3. Astrocyte Secretome Analysis after 6 h Exposure to Exogenous sAPPα

#### 2.3.1. The Spectral Library

In the secretome samples derived from the media of sAPPα-treated astrocytes and the matched controls, a total of 933 proteins and 30,076 peptides were identified using DDA mass spectrometry analysis at a confidence interval of 99% and an FDR of q ≤ 0.01. The identified peptides and proteins were integrated into the spectral library and aligned to the DIA spectra for quantification. A total of 858 proteins and 4948 peptides were used for quantification after passing the FDR threshold for peak matching of q ≤ 0.01 in at least one sample. Twenty-five differentially regulated proteins were identified from 136 peptides (each from more than one peptide) that showed a significant difference (log2(FC) ≤ –0.58 or ≥ 0.58, −log10(*p*-value) ≥ 1.3) in relative abundance between sAPPα-treated astrocytes and the matched controls, as indicated by the orange points in the volcano plots (Figure 5A, Appendix A). An expanded dataset of 93 differentially regulated proteins derived from 133 peptides was quantified in the individual peptide analysis after filtering for *p*-value ≤ 0.05 and fold change (FC) ≥1.5 and ≤0.67 (Figure 5B). Appendix A shows all the proteins in the peptide-level data that were up- and down-regulated, respectively.

#### 2.3.2. Changes in the Astrocyte Secretome at the Protein Level

Functional network analysis of the 25 differentially regulated proteins (Appendix A) revealed a low degree of network connectivity between the proteins with a protein–protein interaction (PPI) enrichment *p*-value of 0.009 and one main cluster of seven protein nodes (APP, GSN, GM2663, DBN1 APLP2, CALM3 and HSP90B1), with APP having the highest betweenness centrality in the whole network. In the pathway enrichment analysis of all the significantly secreted proteins (Appendix A), eight proteins (APP, RPL10, CALM3, DBN1, ATP1A1, HNRNPH2, ADD1 and CDH13) were associated with the GO term ‘Synapse’ under ‘cellular compartment’ (GO:0045202; threshold, 0.67; FDR, 0.0360), six proteins (APP, HSP90B1, DBN1, GSN, PAWR and ADD1) were associated with the biological process term ‘Supramolecular fibre organization’ (GO:0097435; strength, 1.05; FDR, 0.0388) and five proteins (APP, CALM3, ADD1, CDH13 and AP2A1) were associated with the ‘Regulation of endocytosis’. Five proteins in this network had first-level interactions (edges) with APP (GSN, GM2663, DBN1 APLP2, and HSP90b1), of which most, except for GM2663 and APLP2, are involved in supramolecular fibre organization, a GO term encompasses the proteins involved in the process of assembly and disassembly of protein units into fibres, such as actin and tubulin.

#### 2.3.3. Differentially Regulated Secretome Proteins Identified from Individual Peptides

The 93 proteins detected as being significantly differentially regulated are shown in Appendix A, with all of the proteins derived from the individual peptide-level data indicated as up- and down-regulated, respectively. Nineteen of these proteins (APP, BMP1, COL5A2, CST3, CTSD, DBN1, FN1, GALC, GDI2, GOT2, HSP90B1, HSPG2, P4HB, PDIA3, PEBP1, QSOX1, SERPINH1, SPARC and SPARCL1) have previously been reported from the secreted fraction of astrocytes [20,21,62].

STRING network analysis of the differentially regulated secreted proteins revealed a network of associated proteins with three main clusters (Figure 6) enriched in functional pathway terms of (i) the red cluster: supramolecular fibre organization (10 nodes; strength, 1.14; FDR, 2.23 × 10^−5^), synapse (11 nodes; strength, 0.68; FDR, 0.00088) and extracellular matrix organization (nine nodes; strength, 1.3; FDR, 9.92 × 10^−7^); (ii) the green cluster: carboxylic acid metabolic process (nine nodes; strength, 1.19; FDR, 1.65 × 10^−5^) and myelin sheath (four nodes; strength, 1.39; FDR, 0.0163); and (iii) the yellow cluster: ribosome biogenesis (six nodes; strength, 1.45; FDR, 0.00017) and proteasome complex (six nodes; strength, 2.13; FDR, 1.22 × 10^−8^). APP is a central node in the largest cluster (red cluster) with 13 first-order interactions (CFL1, DBN1, GNAO1, NRD1, CALU, PDIA3, HSP90b1, CST3, CTSD, NES, SPARCL1, FN1 and AGRN), mostly to proteins involved in supramolecular fibre organization (CFL1, DBN1, HSP90b1 and CST3) and synapse organization (SPARCL1, AGRN, CFL1 and DBN1), further enriching the functional association that was detected in the protein-level analysis. Further analysis using the SynGO resources [59] identified 28 proteins of the 93 differentially regulated proteins as being related to synaptic functions.

A combination of functional pathway enrichment analysis (STRING interactions and Gene Ontology enrichment analysis (Table 4 and Appendix A) and literature searching revealed 66 proteins of particular interest (~70% of those identified as differentially regulated) as they were associated with at least one of the neuronally related GO terms of neuronal synapse, dendrite, axon, myelin sheath, actin cytoskeleton, extracellular matrix organization, IGF regulation, Alzheimer’s disease KEGG pathway or have a link to APP (Table 5). Some proteins belonged to more than one GO category; for example, 12 proteins were associated with the Cellular Compartments ‘Neuron Projection’ and ‘Synapse’, and two more with these categories and ‘Myelin sheath’ (see Appendix A).

In Table 5, we have listed all of the proteins that were differentially up- or down-regulated in the secretome that were identified under a specific GO term related to neurological function (synapse, neuron projection, myelin sheath, cytoskeleton, ECM organization or IGF regulation) or to APP (sAPPα) and the pathways linked to AD. Most differentially regulated proteins score in more than one category, and approximately 80% of those identified fell into these categories.

### 2.4. Comparison of the Proteome and Secretome Differentially Regulated Proteins

If a protein in the cellular proteome were decreased from exposure to extracellular sAPPα but elevated in the secretome, it suggests that the primary event is stimulated secretion without the enhancement of new synthesis. When both are elevated, it suggests de novo synthesis within the astrocyte, which is reflected in enhanced secretion, and when both are decreased, it suggests that enhanced turnover of the protein in the cell occurs without the de novo synthesis of that protein, resulting in lowered secretion.

Seven proteins were regulated in both the astrocyte cellular proteome and the secretome (Table 6). Three are regulated in the same direction, with one down-regulated (CKB) and two up-regulated (DBN1 and TPM4), and four are down-regulated in the proteome but up-regulated in the secretome (COL5a, ADD, PSMD3 and HNRMP2b1).

## 3. Discussion

### 3.1. sAPPα Modulates the Proteome and Secretome of the Astrocyte

We have identified a group of differentially regulated proteins relevant to neurological function; for example, actin-binding protein Drebrin, extracellular matrix protein SPARC, Myelin sheath protein SOD1, axon guidance protein DPYSL3, proteins associated with endocytosis, such as CALM3, and proteins involved in the regulation of Insulin-like Growth Factor transport and uptake (Reactome mmu381426), providing a focus for the further investigation of their relationship to sAPPα. A STRING functional association network analysis of both the proteome and secretome data sets was enriched with proteins that have links to APP (Figure 3 and Figure 6, respectively) and AD and that is associated with neurological functions (Table 2, Table 3 and Table 5). Intriguingly, several proteins with well-documented roles at the neuronal synapse are enriched in these data, suggesting that these proteins may have similar functions in the astrocyte or, alternatively, are transported to these regions in the neuron from the astrocyte. These findings are consistent with our hypothesis that astrocytes, as well as neurons, respond to sAPPα signaling.

Astrocytes participate in the encoding of memory stimulated after neuronal activity [3], and sAPPα signaling may play a part in this role. The regulation of translation in astrocytes by neuronal activity is also proposed to be a mechanism by which astrocytes modulate nervous system functioning [63,64]. Astrocytes modulate synaptic function by secreting neurotrophic factors, controlling synaptic formation [65] and regulating the concentration of neurotransmitters at the synapses [2]. They also express APP [66] and can generate their own sAPPα extracellularly. Astrocytes have mechanisms for endocytosis [67] and, like neurons, undergo rapid cellular morphology changes in response to chemical or electrical stimulation [68].

When there is an increased extracellular concentration of sAPPα (as simulated in our study), a group of proteins in the astrocyte was modulated that are directly involved with the regulation of the cytoskeleton (control of cell morphology and vesicle dynamics), the regulation of the ECM, the modulation of the myelin sheath (signaling to oligodendrocytes) and activities at the synapse between neurons.

It is known that sAPPα influences learning and memory by modulating the expression of the proteins involved with neurite outgrowth, synaptogenesis and neurotransmission [34,54,55,56,57,69]. Given the close proximity of astrocytes to neurons and their expression of several putative sAPPα receptors [39,48,49,50,51,52,53], sAPPα produced in vivo by neurons or generated extracellularly from the astrocyte itself could influence the molecular biology of astrocytes.

### 3.2. Astrocyte Differentially Regulated Proteins

Of the 271 differentially regulated proteins identified in the whole-cell proteome, 108 (~40%) had links to either APP (15), AD (17) or the GO terms ‘Synapse’ (48), ‘Neuron projection’ (37), ‘Myelin sheath’ (11) or ‘Cytoskeleton’ (57). It should be noted that for many of the regulated proteins identified in these studies, most publications to date relate to studies of their involvement in the neuron only, and so the mechanisms of action of the proteins in the astrocyte itself are yet to be experimentally confirmed. Where the proteins refer specifically to neuronal functions, such as the synapse, we infer that the proteins likely affect the astrocyte itself but may also regulate neuronal function either via the secretome or from a close association of the astrocyte with the neuron through downstream signaling or direct association.

#### 3.2.1. APP- and AD-Related Proteins

Both the proteome and secretome data sets contain numerous differentially regulated proteins with first-order network interactions with APP and associations with AD (Table 3 and Table 5). An example of an APP-related protein of interest was A Disintegrin and Metallo Protease 9 (ADAM9), which is down-regulated, and the net effect of this would be a reduction in the inhibition of ADAM 10, potentially increasing sAPPα production [70]. Proteins associated with the enriched KEGG pathway ‘Alzheimer’s disease’ were of interest given the role of APP in the pathology of the disease, namely the source of neurotoxic peptide Aβ. For example, presenilin (PSEN2) is a gamma secretase involved in the generation of Aβ. Some proteins in the ‘Alzheimer’s disease KEGG pathway’ are risk factors for AD, as identified from variants of the normal genes. PSEN2 is an example of a risk factor and is identified as being down-regulated in our data set. In our study, members of the pathway are both up- and down-regulated and may be beneficial, but the significance of this for brain health is not yet clear.

Other neurological diseases represented by proteins in these data include PD, ALS and HD (Table 1 and Table 4; Appendix A). Alzheimer’s disease is associated with aberrant APP processing in neurons with an over-abundance of Aβ linked to disease pathology [71]. Astrocytes contribute to the pathology of AD, with astrocyte atrophy occurring in the early stage of the disease, leading to disruptions in synaptic connectivity, an imbalance of neurotransmitter homeostasis and neuronal cell death through increased excitotoxicity [72]. Thus, it is interesting that a group of proteins related to these processes is differentially regulated in astrocytes after sAPPα exposure.

#### 3.2.2. Actin and Vesicle Dynamics

Does sAPPα signaling also induce morphology changes in astrocytes? sAPPα is known to cause changes to the actin cytoskeleton in neurons as it can stimulate neurite outgrowth [40,41,42,43], and here we show that 57 proteins associated with the cytoskeleton are differentially regulated in the proteome of astrocytes after sAPPα exposure, and there are 53 proteins specifically associated with plasma membrane cell projection (Table 1 and Appendix A). Although documented to date only in neurons, a group of ‘axon guidance’ proteins (DPYSL3, EPHA4, TUBB2A, COL5A2, SPTAN1, RPS27A AND AP2A1) from the reactome database are featured in the proteome data.

Does sAPPα signaling also modulate vesicle dynamics? Actin dynamics are also integral to the controlled endocytosis and exocytosis of vesicles containing molecules important for synaptic function, with the actin cytoskeleton and associated motors and adaptor proteins essential for the transport and fusion of vesicles at the plasma membrane. In the secretome, at the protein level of analysis, four proteins identified in our study were associated with the GO term ‘Regulation of endocytosis’ (CALM3, ADD1, CDH13 and AP2A1). At the peptide level of analysis, 21 and 46 proteins were associated with the GO term ‘vesicle’ in the secretome and proteome data sets, respectively, with five proteins (HSPD1, ANXA5, QSOX1, FN1 and HNRNPA2B1) in the secretome data specifically associated with the extracellular vesicle compartment (Table 4 and Appendix A). In neurons, vesicle fusion releases neurotransmitters to mediate synaptic transmission and sustain synaptic transmission. The fused vesicles must be recycled and retrieved via endocytosis [73]. In astrocytes, secreted vesicles contain molecules that signal to neurons, for example, to provide neuroprotection [74,75] or to stimulate apoptosis [76].

#### 3.2.3. Myelin Sheath

Does sAPPα signaling influence myelination? Both the proteome and secretome data sets contained groups of proteins that are differentially regulated, which are involved with the myelin sheath, 11 proteins (CKB, WDR1, ATP6V1B2, MDH2, NEFM, SOD1, SEPT2, SLC25A4, CDC42, COX5B and TUFM) in the proteome, and 12 (CKB, CALM3, HSPD1, PDIA3, LDHB, GOT2, ATP1A1, PEBP1, GDI2, SPTAN1, RPS27A and GNAO1) in the secretome. The significance of this requires further investigation; however, astrocytes are known to signal to oligodendrocytes to modulate the myelination process [77,78].

#### 3.2.4. ECM Organization

The secretome data included 34 proteins known to reside in the extracellular space but also eight proteins involved in the organization of the ECM (SPARC, BMP1, P4HB, FN1, COL5A2, CTSD, SERPINH1 and AGRN). During development, astrocytes secrete factors that promote neurite outgrowth [79]. The composition of the ECM is important for maintaining the proper functioning of neuronal synapses [80], and it is a drug target for neurological diseases [81].

#### 3.2.5. sAPPα as a Signaling Molecule

sAPPα (and BDNF) share some common functionalities with Insulin-like Growth Factor 2 (IGF2), and memory retention and LTP can be enhanced and ‘forgetting’ prevented by the addition of endogenous Insulin-like Growth Factor 2 (IGF2) [82,83]. A study analysing astrocyte protein secretion and gene expression in three mouse models of genetic neurodegenerative diseases (Rett, Fragile X and Down syndrome) showed that astrocytes secrete the IGF2 inhibitory protein IGFBP2, causing learning and memory failure [84]. A group of IGF-related proteins (Reactome: mmu 381426) were identified in the secretome data (Table 5). Previous data from our laboratory have shown that sAPPα up-regulates the transcripts of IGF2 and IGFBP2 transiently after 30 min exposure in a human neuroblastoma SH-SY5Y cell culture model [85].

### 3.3. Functions of the Proteins Differentially Regulated in Cellular Proteome and Secretome

Of the proteins differentially regulated in both the proteome and secretome, three proteins (DBN, ADD1 and TPN4) are modulators of morphology or cytoskeletal proteins, while COL5a (collagen) is a constituent of the extracellular matrix and PSMD3 is part of the proteasome. Tropomycin (TPM4) influences the associations between actin and the other actin-binding proteins. Developmentally regulated brain protein, Drebrin (DBN), an actin-binding protein, is of particular interest in that it modulates long-term memory and synaptic plasticity in neurons by influencing synaptic morphology, specifically axonal growth [86,87,88]. The localisation of NMDA receptors (also expressed in hippocampal astrocytes) can be regulated by DBN [89]. In AD patients, DBN expression is reduced [90], and its overexpression in an AD mouse model rescues cognitive deficits and reduces amyloid plaque load [91]. ADD1 belongs to the adducin family of membrane–cytoskeleton-associated proteins, which, in neurons, are constituents of synaptic structures, such as dendritic spines and the growth cones of neurons, where they are involved in the assembly and disassembly of the actin cytoskeleton and responsible for synaptic plasticity and the modulation of synaptic strength [92].

Heterogeneous nuclear ribonucleoproteins (HNRNPs) transport RNAs between the nucleus and the cytoplasm [93]. In neurons, HNRNP-controlled localization-dependent translation of GLUA1 mRNA modulates synaptic AMPA receptor levels [94]. HNRNPA2B1 was down-regulated in the proteome but significantly increased in the secretome, and this protein regulates the selective sorting and secretion of miRNA into extracellular vesicles [95,96]. The differential effects of sAPPα on the proteome and secretome could indicate the stimulation of the secretion of vesicles.

### 3.4. Limitations of the Study

The astrocyte preparation was from the whole-mouse cortex, resulting in multiple astrocyte types; therefore, the data represented here may have given rise to an ‘averaged’ response. The isolated astrocytes in this artificial cell culture system are also lacking input from other glia and neurons during culture, which may influence the effects of sAPPα in vivo. It will be useful to undertake further validation of the selected proteins using a single astrocyte cell type alongside an immunocytochemistry analysis of slice cultures taken from the brain.

The number of regulated proteins with multiple peptide identifications above allowable detection limits was relatively small and may reflect the heterogeneous population of different types of astrocytes. Therefore, we also analysed individual peptide-level data at a relaxed significance threshold, which, in many cases, resulted in a single peptide quantification of an affected protein, albeit with good concordance within the biological and technical replicates. Individual peptide-level data confirmed the protein-level data but identified additional proteins in already-enriched networks and functional pathways.

Although SWATH MS, in theory, captures spectral data for every peptide, the abundance of each peptide and the set confidence limits, as well as the size of the spectral library, the mean data reported in the current study represent a significant sampling but not the complete set of the SDR proteins after exposure to sAPPα in primary mouse astrocytes.

## 4. Materials and Methods

### 4.1. Mouse Primary Astrocyte Cell Culture

Primary astrocytes were derived from the brain tissue of male P0–P2, C57BL/6 mice. The pups were housed at the Hercus-Taieri Resource Unit (HTRU), University of Otago. The experimental protocol was approved by the University of Otago Animal Ethics Committee and conducted in accordance with New Zealand Animal Welfare Legislation. The preparation of the primary astrocyte cultures followed a modified protocol based on [97] and refined by the Hughes Laboratory (University of Otago).

The pups were housed in the dark on warm water bottles until deeply anesthetised by intraperitoneal injection with 0.1 mL of Pentobarbital (Provet NZ Pty Ltd., Auckland, New Zealand) and placed in a Petri dish. Following a loss of consciousness, as determined by slowed/halted respiration, loss of heartbeat and reflexes, the animal was sacrificed by decapitation, and the cortex was isolated and covered with 1 mL of ice-cold Leibovitz (L)15 complete media (Leibovitz L15 (Hyclone, Logan, UT, USA), 1 × pen-strep (Gibco, Billings, MT, USA) and 6 g/L of Glucose (Sigma, Kawasaki, Japan)). After dicing with a sterile scalpel blade, the tissue was transferred to a 15 mL centrifuge tube (Corning, Somerville, MA, USA), and the tissue was allowed to settle. The supernatant was removed and replaced with 1 mL of digestion media (L15 complete media, papain (Worthington, Columbus, OH, USA, 1.2 U/mL) and DNase1 (Invitrogen, Waltham, MA, USA, 1 U/mL) before placing the tube in a MACS-mix rotor (Miltenyi Biotec, Gaithersburg, MD, USA) for 15 min at 37 °C.

The digested tissue was removed from the MACS-mix rotor and allowed to settle in the safety cabinet. The digestion media was aspirated, and 1 mL of trituration solution (L15 complete media, DNase1 (1 U/mL) and 2% (*v*/*v*) B27 (GIBCO, Life technologies, NZ, Carlsbad, CA, USA)) was added to the tube, followed by a 10 min incubation in the MACS-mix rotor at 37 °C. The supernatant was removed and replaced with fresh trituration solution, resuspended, and triturated through a series of fire-polished Pasteur pipettes of decreasing pore size until a cloudy solution with no cell clumps remained. The sample was passed through a 100 µm cell strainer (Falcon) and centrifuged (300× *g*, 5 min). The supernatant was discarded, and the cells were resuspended in Astrocyte Maintenance Media (AMM, DMEM, 10% (*v*/*v*) FBS, 1% (*w*/*v*) pen-strep (Gibco), 1% (*w*/*v*) L-glutamine (Gibco), 2.5 mM glucose, 0.23 mM sodium pyruvate (Gibco), 10 μM L-leucine methyl ester (L-LME, Sigma)) and plated into a poly-L-lysine (PLL, 10 μg/mL)-coated T25 flask. Each pup yielded enough cells for one T25 flask, which was eventually expanded to two 10 cm dishes for treatment and corresponding control for each animal. L-leucine methyl ester (L-LME, 10 μM) was added to inhibit microglial proliferation [98]. Fresh AMM was added to astrocyte cultures every 2–3 d, and the cultures were passaged with TrypLE™ Express enzyme (TrypLE^TM^, GIBCO) and transferred into a T75 flask when reaching 95% confluence. After 14 d in vitro (DIV-14), the astrocytes were shaken for 6 h at 115 rpm at 37 °C to remove any remaining neurons and microglia. The cells were then subject to a final passage and plated into their appropriate vessel for experiments. The astrocytes were cultured for 3 weeks and characterised by immunocytochemistry (Figure 1 and Appendix A).

### 4.2. Immunocytochemical Characterisation of Astrocytes

Primary cell cultures were analysed by immunocytochemistry using antibodies against GFAP and s100b to confirm the presence of astrocytes and CD68 to assess the level of microglia contamination. The cells (5 × 10^4^/cm^2^) were plated on Poly L lysine (Sigma)-coated coverslips and fixed with 300 µL 4% (*v*/*v*) paraformaldehyde (PFA, Sigma) for 15 min at room temperature. PFA was aspirated, and 300 µL of PBS was added to rinse the wells, followed by blocking for 1 h with 300 µL of PBS containing 3% (*v*/*v*) normal goat serum (Invitrogen). Primary antibody, rabbit anti-GFAP (1:1000, Dako Z0334, Santa Clara, CA, USA), mouse anti-s100b (1:500) (Novus Biologicals, NBP1-41373, Englewood, CO, USA) or rat anti-CD68 (1:500) (Serotec, SEMCA1957, Kidlington, UK) was diluted in 350 µL 3% (*v*/*v*) NGS in PBS containing 0.1% (*v*/*v*) Triton-X (BDH) and incubated at 4 °C overnight. After antibody removal, the cells were washed three times with a 10 min incubation of 350 µL PBS, 0.1% (*v*/*v*) Triton-X. Fluorescently labelled secondary antibody Rabbit^448^ (1:1000, Invitrogen A11032), Mouse^594^ (1:1000) (Invitrogen A11029) or Rat^555^ (1:1000) (Invitrogen A21434) was diluted in 300 µL 3% (*v*/*v*) NGS in PBS, 0.1% (*v*/*v*) Triton-X and incubated at room temperature for 1 h. Three PBS-Triton-X washes removed unbound antibodies, and PBS was added to each well. DAPI was used as a nuclear stain. The cells were imaged on a Fluorescent Eclipse Ti2 microscope (Nikon, Tokyo, Japan) attached to an Intensilight C-HGF1 light source (Nikon) and a DS-Qi2 camera (Nikon) linked to a computer with NIS-Elements D imaging software (Nikon). The percentage of microglia expressing CD68 was determined for each culture (four animals) by taking the average of three random images.

### 4.3. Preparation of the sAPPα Protein Used in the Study

An established protocol was used for the expression and purification of human sAPPα^695^ [44]. Briefly, sAPPα was expressed and secreted into serum-free DMEM media (GIBCO) from HEK cells that had been stably transformed with the gene fragment for sAPPα. The protein from 500 mL of the media was concentrated by precipitation with 60% (*w*/*v*) ammonium sulphate, followed by centrifugation at 10,000× *g* for 45 min and resuspended in 20 mM Tris–HCl, pH 7.0. Residual salt was removed by FPLC coupled to a HiTrap^®^ desalting column (Cytiva, Marlborough, MA, USA), and a Heparin Sepharose column (HiTrap^®^ heparin) was used for affinity purification. SDS-PAGE, Western blotting, and BCA assay analysis confirmed the purity and identity and determined the concentration of the purified sAPPα, respectively.

### 4.4. Study Protocol

Primary astrocytes were cultured from four individual P2 mouse pups (n = 4) in 10 cm^2^ cell culture dishes and incubated with either PBS or 1 nM sAPPα for 2 h (proteome study) or 6 h (secretome study). The cells were in serum-containing medium during the 2 h exposure to sAPPα for the proteome study. The 2 h time was chosen for the proteome study to assess the *immediate* effects of exposure to sAPPα, and 6 h for the secretome study to allow time for secretion. The time point rationale was based on previous studies using neural cells, tissue slices and whole animals that measure changes in gene expression after a stimulus (including sAPPα) [55,56,57,99]. The isolation of the proteins from the secreted fraction has been previously described [18,62] and was modified for this study. Prior to the addition of sAPPα, the cells were rinsed thoroughly (6 times) with pre-warmed PBS to remove any serum proteins. Serum-free cell culture media (7 mL) containing 1 nM sAPPα or the equivalent volume of 0.1 × PBS (vehicle) was added to the cells, which were returned to the incubator for 6 h. The cell culture media was harvested and centrifuged at 300× *g* for 5 min and 1000× *g* for 10 min followed by 20,000× *g* for 25 min. After each centrifugation, the supernatant was transferred to a fresh 2 mL microcentrifuge tube. Twelve successive centrifugation steps concentrated 6 mL of the media in 500 μL lots on a 10 kDa cut-off ultra-concentrating filter (Amicon, EMD Millipore), down to 60 μL (100-fold concentration) and all of the centrifugation was carried out in a temperature-controlled microcentrifuge (Microfuge 20R centrifuge, Beckman Coulter) at 20 °C. The samples were recovered by inverting the filter unit and centrifuging the retained liquid into a fresh Eppendorf tube at 2000× *g* for 2 min. Twenty μL of 50 mM triethylammonium bicarbonate (TEAB) and 1% (*w*/*v*) SDS in water was added to the filter and vortexed gently to rinse and elute any remaining protein. This 20 μL was pooled with the recovered retained liquid.

For the analysis of the cellular proteome, after 2 h of treatment with 1 nM sAPPα or PBS, the cells were harvested with TrypLE™ enzymatic digestion, rinsed with PBS, pelleted by centrifugation, snap frozen in an ethanol/dry ice bath and stored at −80 °C. To extract the protein fraction, the media or frozen cell pellets were thawed in 200 μL of solubilisation buffer (500 mM (TEAB), 1 mM of phenylmethylsulfonyl fluoride (PMSF), 1 mM of ethylenediaminetetraacetic acid (EDTA), 1 mM of ethylene-bis(oxyethylenenitrilo)tetraacetic acid (EGTA), 0.1% (*w*/*v*) sodium dodecyl sulphate (SDS) and 1% (*w*/*v*) sodium deoxycholate (SDC). To aid the lysis and digestion of the whole-cell proteome, the cell suspension was homogenised with twenty grinds of a pestle (sample grinding kit, GE Healthcare). The pestle was then rinsed with 200 μL of urea solution (8 M urea in aqueous 500 mM TEAB). The homogenate was vortexed for 10 s and sonicated for 1 min, and then supplemented with 100 U of benzonase. After incubation at 37 °C for 30 min, the cell lysate was centrifuged at 16,000× *g* for 30 min at 20 °C to remove insoluble material and cell debris. The supernatant was removed, carefully avoiding any top lipid layers and insoluble pellets. The recovered supernatant was further processed as the soluble protein fraction. The soluble protein fractions from the cell lysates and the media from the secretome fraction were each transferred to a 10 kDa molecular weight cut-off centrifugal filter unit (Amicon Ultracell—EMD Millipore, Burlington, MA, USA) and further processed following the Filter-Aided Sample Preparation (FASP) method [100]. In brief, detergents were depleted by washing the samples with 8 M of urea in aqueous 100 mM TEAB, followed by the reduction and alkylation of cysteine thiols with 5 mM tris (2-carboxyethyl) phosphine (TCEP) and 10 mM iodoacetamide (IAA) in aqueous 100 mM TEAB, respectively, and a final washing in aqueous 100 mM TEAB.

The protein content of each sample (secretome and proteome) was quantified with the Bradford protein quantitation assay (Biorad protein assay, Biorad, Hercules, CA, USA) and then supplemented with trypsin (Sequencing Grade Promega) in a trypsin-to-protein ratio of 1 to 20 (*w*/*w*) for overnight digestion at 37 °C. After 14 h, the digests were boosted with an additional amount of trypsin in a trypsin-to-protein ratio of 1 to 40 (*w*/*w*) and incubated for another 4 h at 37 °C. After digestion, the peptides were purified and concentrated by solid phase extraction (SPE) on Sep-Pac C18 cartridges (Waters, Milford, MA, USA) and dried using a centrifugal vacuum concentrator.

### 4.5. Protein Identification and Quantification by SWATH-MS

A comprehensive spectral library containing the peptide spectra of all of the identified proteins from all of the samples was generated through data-dependent acquisition (DDA) mass spectrometry of a pooled sample containing an equal amount of each sample. To achieve greater protein identification, the pooled sample was subjected to peptide pre-fractionation by Off-Gel isoelectric focusing (OG-IEF). Therefore, the peptides were first purified through solid phase extraction on Sep-Pak Plus Light C18 cartridges (Waters) and then fractionated into 12 fractions using OG-IEF along a linear pH gradient from 4 to 10 using a 3000 OFFGEL Fractionator (Agilent Technologies, Santa Clara, CA, USA) according to the manufacturer’s protocol. Each fraction was then analysed using DDA mass spectrometry on a 5600+ Triple Time-Of-Flight (TOF) mass spectrometer coupled to an Eksigent “ekspert nanoLC 415” uHPLC system (AB Sciex, Framingham, MA, USA), as previously described (Sweetman, 2020) [101]. For peptide/protein identification, the raw data files of all fractions were searched against the human reference sequence database (downloaded from https://www.ncbi.nlm.nih.gov/ on 29 March 2019), which contains 87,570 sequence entries, using the Protein Piolet software (version 4.5). Trypsin, carboxymethylcysteine and biological modifications were selected for a thorough search setup.

Protein quantification was achieved through data-independent acquisition (DIA) mass spectrometry using the SWATH-MS workflow according to the details described previously [101]. Each sample was analysed in four technical replicates using the same LC gradient as that used for the DDA analysis.

### 4.6. Data Analysis and Statistics

The spectral library was built using the SWATH application (version 2.0) embedded into PeakView software (version 2.2, AB Sciex), applying the criteria outlined in [101]. DIA spectral data were aligned to the library spectra at a false discovery threshold for peak picking of q = 0.01 in at least one sample. The peak area under the curve (AUC) was then exported to MarkerView software (version 1.2, AB Sciex) for further statistical analysis. The median value of the AUC of the four technical replicates was used to calculate the mean between the biological replicates of sAPPα-treated or control samples for each peptide and protein. The mean peptide/protein values were then compared between sAPPα-treated and control using a 2-tailed Student’s t-test. The proteins were identified as being SDR if they were up- or down-regulated by ≥1.5-fold (Log2 fold change ≥ 0.58 or ≤−0.58) with a *p*-value of ≤0.05 (−log10(*p*-value) ≤ 1.3).

Regulated proteins were analysed through functional association network analysis using the STRING database tool (https://string-db.org/) (accessed on 9 October 2022), which identifies the protein–protein associations previously documented in databases or mentioned together in the literature and highlights the groups of proteins that are statistically enriched for their associations with Gene Ontology (GO) terms, such as cellular localisation or biological function or specific molecular pathways or diseases [102,103]. For both the up- and down-regulated proteins APP, the protein that sAPPα is derived from, was added to the network search. To specifically search for synapse functions, regulated proteins were further analysed using the Synaptic Gene Ontology (SynGO) resources (https://www.syngoportal.org/) (accessed on 9 October 2022).

The comparison of sAPPα-treated and control samples was performed using protein-level data, i.e., the average AUCs of the peptides assigned to the same protein and at the level of individual peptide AUCs. In a small number of cases where two or more different proteins were identified from the same peptide sequence, the entries were omitted, as the identity of the protein could not be determined unequivocally. Where >1 isoform of the same protein was highlighted, one isoform was carried forward for further analysis, as the identity of the specific isoform could not be determined.

The individual peptide-level data has been used to find further potentially regulated proteins that are functionally associated with the targets identified on the protein level. However, where the proteins that have been identified as regulated on the individual peptide level are based on single-peptide quantification, those of interest would require further consideration.

## 5. Conclusions

In this study, we have tested the hypothesis that sAPPα regulates the neurologically relevant activities of astrocytes. Previously, our studies have shown that sAPPα is active in regulating neurological activity globally in live rodents [30], in hippocampal tissue slices [55] and in cultured neurons derived from stem cells [57]. Here, we have shown that sAPPα changes the astrocyte proteome, and thereby, the proteins in the secretome, and many of the identified proteins have neurologically related functions. The identification of proteins affected by acute exposure of astrocytes to sAPPα and changes in the array of proteins that are secreted by astrocytes indicates that sAPPα acts as a signaling molecule to the astrocyte. A consequence for neurons is inferred to be the mobilisation of the necessary tools and materials required for modifying or making new synaptic connections or potentiating existing synapses. It is consistent with the premise that bidirectional communication between astrocytes and neurons is necessary for controlled neurotransmission and that astrocytes participate in the encoding of memory, stimulated after neuronal activity [3,8]. The proteins identified in this study include a group of proteins involved with alterations to cell morphology and the modulation of vesicle dynamics and that is associated with transmission at the tetrapartite synapse. These proteins form candidates for more focused studies on their functional significance and for co-culture studies of astrocytes with neurons.

Previously, we have shown the potential use of sAPPα as a corrective therapy in compromised live animals through the infusion of the protein into rodent brains that restored memory of spatial learning [30], novel object recognition [31], and promoted long-term Potentiation—a model of mammalian memory (whether a full-length protein or a functionally active peptide RER derived from it) [104]. The long-term induction of neurologically important neuroprotective genes and pathways resulted from the expression of the sAPPα gene fragment in the brain, delivered and expressed in a lentivirus vector [56]. These studies implied that sAPPα activates a diverse range of neurologically important genes and proteins to achieve a final output function. The study of the human neuron in isolation [57] and this current study using astrocyte have identified many of the candidate genes and proteins at the molecular level that sAPPα is able to modulate. It further reinforces the promise of sAPPα and its derivatives for development as a potential therapeutic option for the treatment of Alzheimer’s disease.

## Figures and Tables

**Figure 1 ijms-24-07165-f001:**
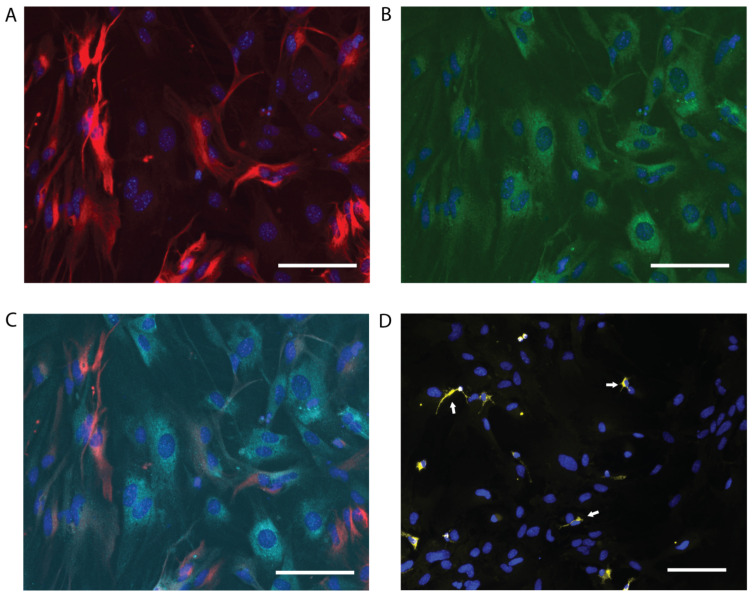
Primary cell cultures are enriched with astrocytes. Representative images from immunocytochemical analysis of DIV21 mouse primary astrocyte cultures using astrocytic markers (GFAP and s100b) and microglial marker (CD68). (**A**) anti-GFAP (red); (**B**) anti-s100b (green); (**C**) co-staining with anti-GFAP (red), anti-s100b (green); (**D**) anti-CD68 (yellow). Microglia are indicated by white arrows. In all images, the nuclei were stained with DAPI (blue). Scale bars: 50 mm.

**Figure 2 ijms-24-07165-f002:**
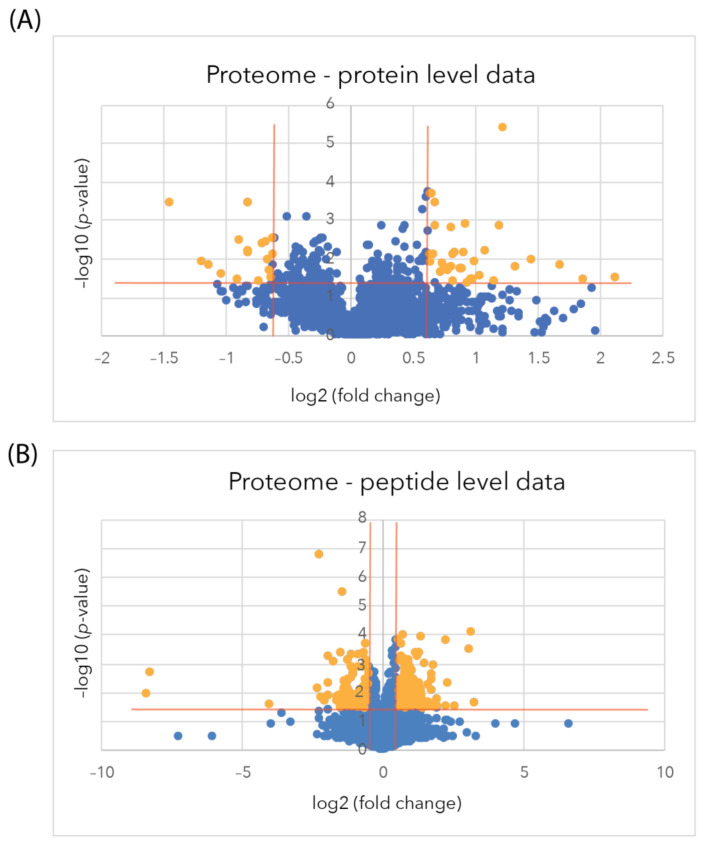
Astrocyte proteome volcano plots after sAPPα treatment. (**A**). Protein level data (**B**). Peptide-level data. Volcano plots show the −log10 *p*-value (y-axis) against Log2 fold change (x-axis) at 2 h to highlight any down-regulated (orange points, top left) and up-regulated proteins (orange points, top right), which meet the criteria for selection of a fold change minimum of 1.5 (log2 fold change ≤ −0.58 or ≥0.58) with a *p*-value of ≤0.05 (−log10 (*p*-value) ≤ 1.3). Blue points represent proteins where there has been no significant change to meet the criteria.

**Figure 3 ijms-24-07165-f003:**
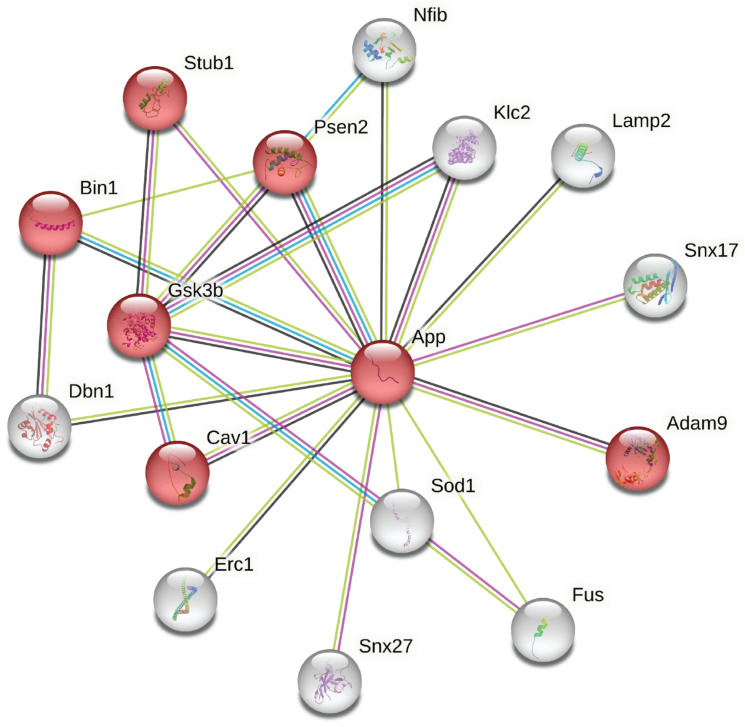
First-order interaction partners of APP (proteome, peptide-level data). Differentially regulated proteins that are functionally associated with APP according to a STRING functional network analysis. The nodes highlighted in red are involved in the regulation of proteolysis and protein catabolic processes. The symbols are designated in Table 3.

**Figure 4 ijms-24-07165-f004:**
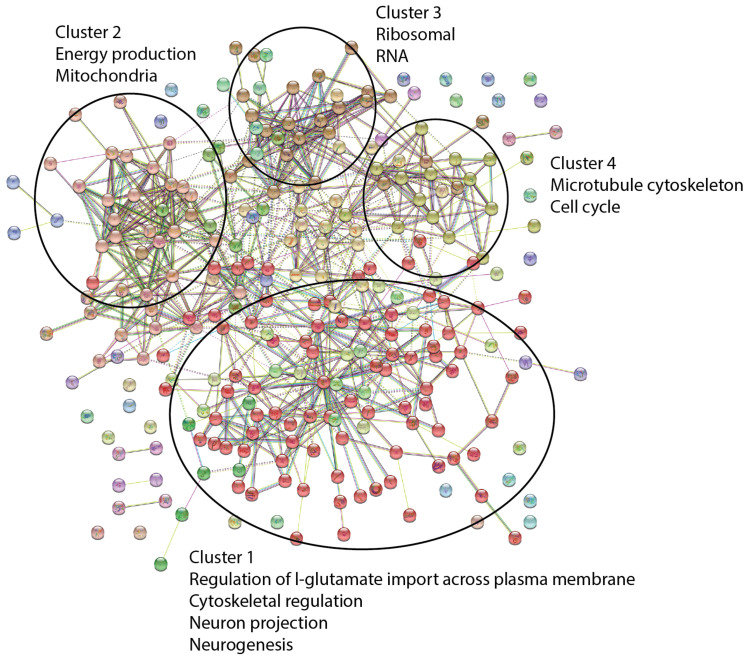
MCL Cluster Analysis of peptide-level proteome data. A selection of enriched GO terms is highlighted. STRING network of differentially expressed proteins was associated with enriched GO terms (coloured). Proteins (nodes) are coloured according to the different clusters they were assigned to after applying the Markov Clustering Algorithm (MCL) (Brohée and van Helden, 2006) [60].

**Figure 5 ijms-24-07165-f005:**
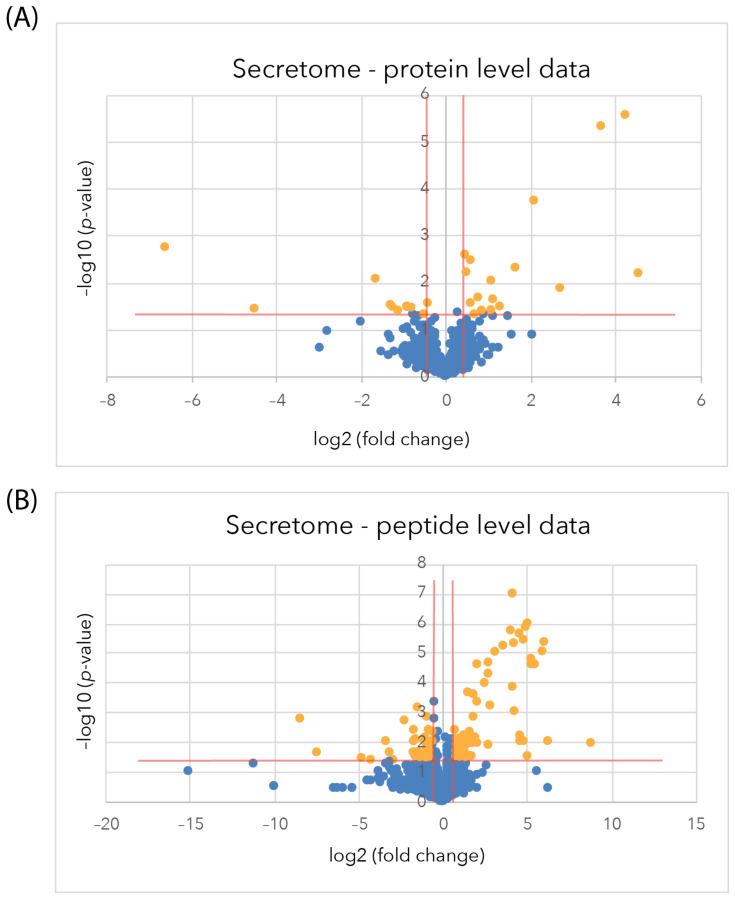
Volcano plots of the proteins identified in the astrocyte secretome from sAPPα-treated cells and matched untreated controls. (**A**). Protein level (**B**). Peptide-level volcano plots the −log10 *p*-value (y-axis) against log2 fold change (x-axis) at 6 h to highlight any down-regulated (left box-orange points) and up-regulated (right box-orange points) proteins, which meet the criteria for selection of a minimum of 1.5-fold change (log2 fold change ≤ −0.58 or ≥0.58) with a *p*-value of ≤0.05 (−log10 (*p*-value) ≤ 1.3). Blue points represent proteins where there has been no significant change to meet the criteria.

**Figure 6 ijms-24-07165-f006:**
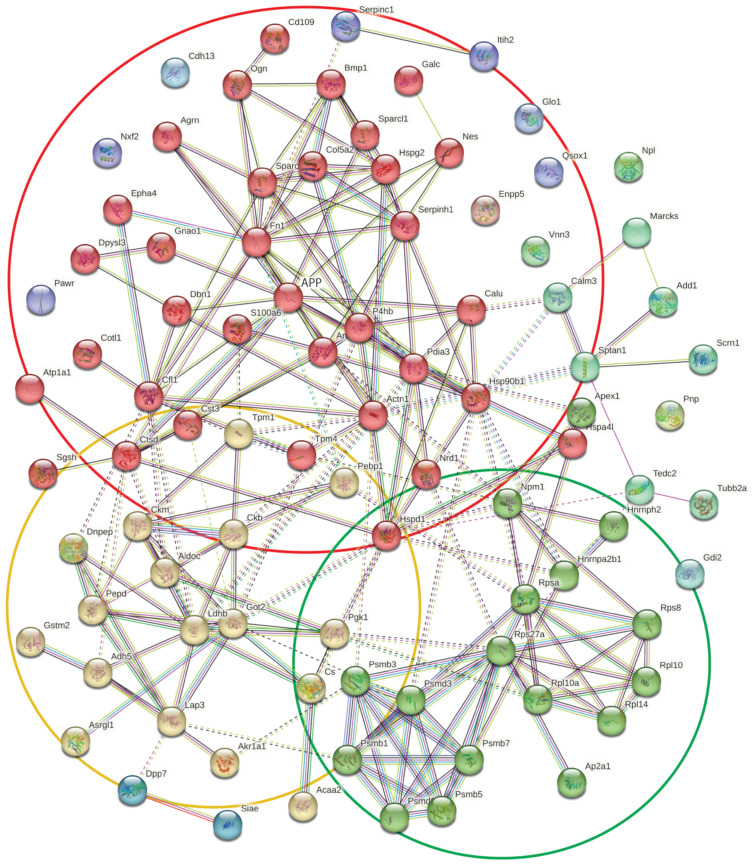
STRING functional network analysis of differentially secreted proteins (peptide-level data) from primary astrocytes after exposure to 1 nM sAPPα for 6 h. The network shows three main clusters, which are enriched with the following functional pathway terms. Red cluster: supramolecular fibre organization (for example, actin), synapse and extracellular matrix organization. Green cluster: carboxylic acid metabolic process and myelin sheath. Yellow cluster: ribosome biogenesis and proteasome complex. APP forms the central node in the largest cluster (red cluster), with 13 first-order edges within the cluster.

**Table 1 ijms-24-07165-t001:** Enriched GO terms. A selection of enriched categories linked to neurological function. GO component: Gene Ontology Cellular Component; GO Process: Gene Ontology Biological Process. Abbreviations: OGC, overall gene count; BGC, background gene count; FDR, false discovery rate. Refer to Appendix A for related proteins.

Category	GO Term ID	GO Term Description	OGC	BGC	Strength	FDR
GO Component	GO:0045202	Synapse	48	1492	0.42	1.11 × 10^−7^
	GO:0030054	Cell junction	58	2050	0.37	1.50 × 10^−7^
	GO:0030424	Axon	26	791	0.43	0.00031
	GO:0005856	Cytoskeleton	57	2060	0.36	3.72 × 10^−7^
	GO:0015630	Microtubule cytoskeleton	32	1190	0.35	0.0011
	GO:0098794	Postsynapse	28	761	0.48	1.78 × 10^−5^
	GO:0030135	Coated vesicle	12	210	0.67	0.00074
	GO:0031982	Vesicle	46	2007	0.28	0.00097
	GO:0098978	Glutamatergic synapse	19	501	0.5	0.00078
	GO:0099572	Postsynaptic specialization	17	449	0.5	0.0018
	GO:0043209	Myelin sheath	11	212	0.63	0.0026
	GO:0043005	Neuron projection	37	1583	0.29	0.0033
	GO:0120025	Plasma-membrane-bounded cell projection	53	2387	0.26	0.00058
	GO:0043197	Dendritic spine	10	212	0.59	0.0083
	GO:0098984	Neuron-to-neuron synapse	15	446	0.44	0.0097
	GO:0014069	Postsynaptic density	14	414	0.45	0.0128
	GO:0099571	Postsynaptic cytoskeleton	3	17	1.16	0.0254
	GO:0030425	Dendrite	19	753	0.32	0.0348
GO Process	GO:0007010	Cytoskeleton organization	33	1064	0.41	0.00026
	GO:0007015	Actin filament organization	14	243	0.68	0.0006
	GO:0030036	Actin cytoskeleton organization	19	496	0.5	0.0025
	GO:0030029	Actin filament-based process	20	551	0.48	0.0031
	GO:0097435	Supramolecular fibre organization	22	469	0.59	3.82 × 10^−5^
	GO:0008064	Regulation of actin polymerization or depolymerization	11	181	0.7	0.0033
	GO:0030705	Cytoskeleton-dependent intracellular transport	11	186	0.69	0.004
	GO:0110053	Regulation of actin filament organization	12	266	0.57	0.0154
	GO:0051493	Regulation of cytoskeleton organization	18	541	0.44	0.0161
	GO:0010970	Transport along microtubule	9	155	0.68	0.0175
	GO:1902683	Regulation of receptor localization to synapse	5	25	1.22	0.0039
KEGG	mmu05014	Amyotrophic lateral sclerosis	20	364	0.66	7.34 × 10^−6^
	mmu05012	Parkinson disease	16	239	0.74	9.20 × 10^−6^
	mmu05016	Huntington disease	17	296	0.68	2.15 × 10^−5^
	mmu05010	Alzheimer disease	18	359	0.62	5.02 × 10^−5^
Reactome	MMU-195721	Signaling by WNT	13	230	0.67	0.0013
Monarch MPO	MP:0010768	Mortality/aging	124	5549	0.27	2.04 × 10^−10^

**Table 2 ijms-24-07165-t002:** Differentially regulated proteins associated with APP. A STRING network analysis assigns scores relating to the strength of evidence that two proteins are functionally associated, derived from either co-expression data, an experimentally determined interaction, referred to within the same publication, or together in a database. The combined score (>0.4) is indicative of significance, and the interaction was confirmed by a literature search.

Node1	Node2	Coexpression	Experimentally Determined Interaction	Database Annotated	Automated Textmining	Combined Score
APP	CAV1	0.098	0.091	0	0.371	0.439
APP	SNX27	0	0.13	0	0.4	0.455
APP	FUS	0	0	0	0.474	0.474
APP	GSK3	0.061	0	0	0.468	0.479
APP	DBN1	0.116	0	0	0.528	0.565
APP	STUB1	0	0.439	0	0.317	0.6
APP	ADAM9	0.076	0.053	0	0.61	0.629
APP	ERC1	0.062	0	0	0.679	0.686
APP	SOD1	0	0	0	0.72	0.72
APP	SNX17	0	0.061	0	0.72	0.726
APP	BIN1	0.098	0	0.54	0.681	0.856
APP	GSK3B	0.056	0.427	0	0.79	0.876
APP	KLC1	0.118	0.402	0	0.905	0.946
APP	NFIB	0.062	0	0	0.962	0.963
APP	PSEN2	0.048	0.623	0.502	0.933	0.986

**Table 3 ijms-24-07165-t003:** A selection of neurologically significant astrocyte proteins regulated by sAPPα derived from the proteome peptide-level data. Functional pathway analysis using the STRING database highlighted proteins that were associated with (i) a first-level interaction with APP, (ii) GO terms with neurological function (synapse, neuron projection, myelin sheath and cytoskeleton) and (iii) the Alzheimer’s disease KEGG pathway. Abbreviation: FC, fold change. The 

 indicates the protein was classified to be in that category.

(A) Up-Regulated
Symbol	Protein Name	FC	*p*-Value	APP Related	Synapse	Neuron Projection	Myelin Sheath	Cytoskeleton	Alzheimer’s Disease
DBN1	Drebrin 1	9.51	2.30 × 10^−2^						
CYC1	Cytochrome c-1	8.31	3.57 × 10^−4^						
ERC1	ELKS/Rab6-interacting/CAST family member 1	3.82	3.41 × 10^−2^						
RPS7	Ribosomal protein S7	3.47	3.29 × 10^−2^						
RPL26	60S ribosomal protein L26	2.86	2.93 × 10^−2^						
ADNP	Activity-dependent neuroprotective protein	2.67	3.89 × 10^−2^						
TBCA	Tubulin cofactor A	2.65	2.66 × 10^−2^						
MYL6	Myosin light polypeptide 6	2.64	1.21 × 10^−2^						
XRN1	5′-3′ exoribonuclease 1	2.62	9.61 × 10^−3^						
PRC1	Protein regulator of cytokinesis 1	2.54	5.44 × 10^−3^						
WASL	WASP-like actin-nucleation-promoting factor	2.35	3.04 × 10^−2^						
PRKAA1	Protein kinase, AMP-activated, alpha 1 catalytic subunit	2.34	2.13 × 10^−2^						
SOD1	Superoxide dismutase 1, soluble	2.33	6.13 × 10^−4^						
NDUFS8	NADH:ubiquinone oxidoreductase core subunit S8	2.27	2.02 × 10^−2^						
BRK1	BRICK1, SCAR/WAVE actin-nucleating complex subunit	2.25	4.68 × 10^−4^						
TMOD3	Tropomodulin 3	2.22	3.60 × 10^−3^						
TACC3	Transforming, acidic coiled-coil containing protein 3	2.16	2.57 × 10^−2^						
RIF1	Telomere-associated protein RIF1	2.09	6.85 × 10^−3^						
Erbin	Erbb2 interacting protein/LAP2	2.04	4.71 × 10^−2^						
HYPK	Huntingtin interacting protein K	2.03	1.45 × 10^−2^						
PSMD4	Proteasome 26S subunit, non-ATPase, 4	2.02	5.66 × 10^−3^						
SEPTIN2	Septin-2	1.99	1.53 × 10^−2^						
MPP1	Membrane protein, palmitoylated	1.98	2.34 × 10^−3^						
FNTA	Farnesyltransferase, CAAX box, alpha	1.98	3.18 × 10^−3^						
PLEKHA7	Pleckstrin homology domain containing family A member 7	1.95	8.09 × 10^−3^						
SRI	Sorcin	1.94	4.84 × 10^−2^						
NEXN	Nexilin	1.92	3.68 × 10^−2^						
KLC1	Kinesin light chain 1	1.89	2.35 × 10^−2^						
EPB4.1	Erythrocyte membrane protein band 4.1	1.89	4.72 × 10^−2^						
RPS19	40S ribosomal protein S19	1.87	3.56 × 10^−3^						
WDR1	WD repeat domain 1	1.85	2.38 × 10^−2^						
SNX17	Sorting nexin 17	1.78	1.62 × 10^−3^						
COX6B1	Cytochrome c oxidase, subunit 6B1	1.77	3.64 × 10^−2^						
MYO1E	Myosin IE	1.77	3.65 × 10^−2^						
SEPTIN11	Septin 11	1.74	3.61 × 10^−2^						
HNRNPC	Heterogeneous nuclear ribonucleoprotein C	1.74	2.84 × 10^−2^						
MTPN	Myotrophin	1.74	1.76 × 10^−2^						
VSP4B	Vacuolar protein sorting 4B	1.74	3.73 × 10^−2^						
DCTN2	Dynactin 2	1.73	4.21 × 10^−3^						
RPL17	Ribosomal protein L17	1.73	2.74 × 10^−4^						
YWHAH	Tryptophan 5-monooxygenase activation protein	1.71	1.25 × 10^−2^						
AP3B1	Adaptor-related protein complex 3, beta 1 subunit	1.71	2.06 × 10^−2^						
STK39	STE20/SPS1-related proline-alanine-rich protein kinase	1.70	4.09 × 10^−2^						
ARL6IP5	ADP-ribosylation factor-like 6 interacting protein 5	1.68	4.25 × 10^−2^						
ITGB5	Integrin beta 5	1.68	4.91 × 10^−3^						
PPP2CA	Protein phosphatase 2 (formerly 2A), catalytic subunit	1.67	1.06 × 10^−2^						
ELAVL1	ELAV (embryonic lethal, abnormal vision)-like 1	1.65	4.34 × 10^−2^						
NEFM	Neurofilament medium polypeptide	1.65	2.08 × 10^−2^						
TBCEL	Tubulin-specific chaperone cofactor E-like protein	1.65	2.47 × 10^−2^						
TWF1	Twinfilin actin-binding protein 1	1.63	2.01 × 10^−2^						
BIN1	Bridging integrator 1	1.61	4.03 × 10^−2^						
TUFM	Tu translation elongation factor, mitochondrial	1.60	2.22 × 10^−2^						
MCM3	Minichromosome maintenance complex component 3	1.59	1.39 × 10^−2^						
SH3GLB1	SH3-domain GRB2-like B1 (endophilin)	1.59	2.92 × 10^−2^						
TPM4	Tropomyosin 4	1.59	2.61 × 10^−2^						
CDC42	Cell division cycle 42	1.59	5.83 × 10^−4^						
CLTA	Cathrin light chain A	1.59	1.32 × 10^−2^						
NPM3	Nucleoplasmin 3	1.59	2.29 × 10^−4^						
RPLP2	Ribosomal protein, large P2	1.59	4.35 × 10^−2^						
RPL29	60S ribosomal protein L29	1.58	1.81 × 10^−2^						
PSMD2	proteasome 26S subunit, non-ATPase, 2	1.58	9.68 × 10^−3^						
DDAH2	N(G),N(G)-dimethylarginine dimethylaminohydrolase 2	1.58	3.30 × 10^−2^						
NPTN	Neuroplastin	1.56	9.91 × 10^−3^						
KPNA2	Karyopherin (importin) alpha 2	1.54	4.27 × 10^−2^						
MDH2	Malate dehydrogenase 2, NAD (mitochondrial)	1.52	8.93 × 10^−3^						
MAP2K1	Mitogen-activated protein kinase kinase 1	1.52	4.53 × 10^−2^						
**(B) Down-regulated**
**Symbol**	**Protein Name**	**FC**	***p*-Value**	**APP Related**	**Synapse**	**Neuron Projection**	**Myelin Sheath**	**Cytoskeleton**	**Alzheimer’s Disease**
RPS6KA1	Ribosomal protein S6 kinase polypeptide 1	0.003	1.16 × 10^−2^						
GPC4	Glypican 4	0.003	2.21 × 10^−3^						
SNX27	Sorting nexin family member 27	0.20	7.92 × 10^−3^						
PCNA	Proliferating cell nuclear antigen	0.22	1.77 × 10^−7^						
NDUFB9	NADH:ubiquinone oxidoreductase subunit B9	0.27	4.10 × 10^−2^						
NEK9	NIMA (never in mitosis gene a)-related expressed kinase 9	0.27	5.78 × 10^−4^						
HNRNPA2B1	Heterogeneous nuclear ribonucleoproteins A2/B1	0.30	8.82 × 10^−4^						
FARP1	FERM, RhoGEF (Arhgef) and pleckstrin domain protein 1	0.33	1.65 × 10^−2^						
GSK3	Glycogen synthase kinase-3	0.37	3.43 × 10^−6^						
GSK3	Lysosomal-associated membrane protein 2	0.38	2.43 × 10^−2^						
NDUFV1	NADH:ubiquinone oxidoreductase core subunit V1	0.38	2.03 × 10^−2^						
STOML2	Stomatin (Epb7.2)-like 2	0.40	8.65 × 10^−3^						
GLS	Glutaminase	0.41	1.87 × 10^−2^						
STUB1	STIP1 homology and U-Box containing protein 1	0.42	1.09 × 10^−2^						
DARS	Aspartyl-tRNA synthetase	0.42	3.17 × 10^−2^						
NDUFS2	NADH:ubiquinone oxidoreductase core subunit S2	0.42	8.08 × 10^−4^						
COPG2	Coatomer protein complex, subunit gamma 2	0.43	1.53 × 10^−2^						
ENAH	Protein-enabled homolog	0.43	1.59 × 10^−2^						
PSEN2	Presenilin 2	0.44	2.04 × 10^−2^						
ND4	NADH dehydrogenase 4	0.46	1.49 × 10^−3^						
GLRX5	Glutaredoxin 5	0.47	3.15 × 10^−2^						
LAMC1	Laminin subunit gamma-1 precursor	0.48	1.35 × 10^−2^						
NFIB	Nuclear factor I/B	0.49	2.03 × 10^−3^						
ATP5D	ATP synthase, H+ transporting, mitochondrial F1 complex, delta subunit	0.49	3.26 × 10^−2^						
NDRG2	Protein NDRG2, cytoplasmic protein Ndr1	0.52	1.64 × 10^−2^						
DMD	Dystrophin, muscular dystrophy	0.56	1.33 × 10^−2^						
COX5B	Cytochrome c oxidase subunit 5B	0.57	1.45 × 10^−2^						
FUS	Fused in sarcoma	0.58	2.30 × 10^−2^						
PSMD3	Proteasome 26S subunit, non-ATPase, 3	0.59	8.54 × 10^−3^						
LIMA1	LIM domain and actin-binding 1	0.61	3.95 × 10^−2^						
COX7A2	Cytochrome c oxidase subunit 7A2	0.61	2.63 × 10^−2^						
ADAM9	a disintegrin and metallopeptidase domain 9	0.62	3.11 × 10^−2^						
ITSN1	Intersectin 1 (SH3 domain protein 1A)	0.62	2.74 × 10^−2^						
RPL18A	Ribosomal protein L18A	0.63	3.74 × 10^−2^						
CAV1	Caveolin 1, caveolae protein	0.64	1.70 × 10^−2^						
CKB	Creatine kinase	0.65	1.48 × 10^−3^						
ARHGAP39	rho GTPase-activating protein 39	0.65	3.28 × 10^−2^						
PHB2	Prohibitin 2	0.66	2.09 × 10^−2^						
ATP6V1B2	ATPase, H+ transporting, lysosomal V1 subunit B2	0.66	7.47 × 10^−3^						
NEDD1	Neural precursor cell-expressed, developmentally down-regulated gene 1	0.66	7.56 × 10^−3^						
ADD1	Adducin 1 (alpha)	0.66	3.22 × 10^−2^						
SLC25A4	Solute carrier family 25	0.67	3.22 × 10^−2^						

**Table 4 ijms-24-07165-t004:** Enriched GO terms (secreted proteins, peptide-level data). A selection of significantly enriched GO terms related to the proteins that were secreted from astrocytes after sAPPα exposure. GO component, Gene Ontology Cellular Component; GO Process, Gene Ontology Biological Process. Abbreviations: OGC, overall gene count; BGC, background gene count; FDR, false discovery rate. Refer to Appendix A for related proteins and a full list of enriched GO terms.

Category	GO Term ID	GO Term Description	OGC	BGC	Strength	FDR
GO Process	GO:0050807	Regulation of synapse organization	7	275	0.79	0.0278
	GO:0050808	Synapse organization	7	303	0.75	0.0412
	GO:0030036	Actin cytoskeleton organization	11	496	0.73	0.0038
	GO:0007010	Cytoskeleton organization	14	1064	0.5	0.0241
	GO:0034330	Cell junction organization	9	481	0.66	0.0304
GO Component	GO:0005576	Extracellular region	34	2229	0.57	9.14 × 10^−9^
	GO:0005615	Extracellular space	24	1424	0.61	7.45 × 10^−7^
	GO:0043209	Myelin sheath	12	212	1.14	5.67 × 10^−8^
	GO:0045202	Synapse	24	1492	0.59	1.57 × 10^−6^
	GO:0098794	Postsynapse	12	761	0.58	0.0034
	GO:0098978	Glutamatergic synapse	9	501	0.64	0.0085
	GO:0030424	Axon	13	791	0.6	0.0015
	GO:0043005	Neuron projection	19	1583	0.46	0.0015
	GO:0030426	Growth cone	7	224	0.88	0.0024
	GO:0150034	Distal axon	9	399	0.74	0.0024
	GO:0031982	Vesicle	21	2007	0.4	0.0028
COMPARTMENTS	GOCC:1903561	Extracellular vesicle	5	93	1.11	0.0052
	GO:0043197	Dendritic spine	6	212	0.84	0.0089
	GO:0044295	Axonal growth cone	3	42	1.24	0.0194
	GOCC:0120025	Plasma-membrane-bounded cell projection	17	1774	0.37	0.0364
	GO:0014069	Postsynaptic density	7	414	0.61	0.0335
KEGG	mmu00330	Arginine and proline metabolism	4	53	1.26	0.0033
	mmu05010	Alzheimer disease	8	359	0.73	0.0043
	mmu05014	Amyotrophic lateral sclerosis	9	364	0.78	0.0015
	mmu05016	Huntington disease	8	296	0.82	0.0017
	mmu05017	Spinocerebellar ataxia	6	140	1.02	0.0017
	mmu05012	Parkinson disease	9	239	0.96	0.00014
Reactome	MMU-381426	Regulation of IGF transport and uptake by Insulin-like Growth Factor Binding Proteins (IGFBPs)	10	122	1.3	6.95 × 10^−8^
	MMU-422475	Axon guidance	7	278	0.79	0.0021
	MMU-1474228	Degradation of the extracellular matrix	6	144	1	0.00047
	MMU-1474244	Extracellular matrix organization	11	295	0.96	1.30 × 10^−6^
	MMU-5687128	MAPK6/MAPK4 signaling	7	74	1.36	1.11 × 10^−6^
	MMU-3858494	Beta-catenin-independent WNT signaling	10	127	1.28	8.04 × 10^−8^
	MMU-5676590	NIK → noncanonical NF-kB signaling	7	57	1.47	5.37 × 10^−7^
Monarch MPO	MP:0010768	Mortality/aging	44	5549	0.28	0.009

**Table 5 ijms-24-07165-t005:** Regulated secreted proteins (peptide-level data). Summary of functional pathway analysis using the STRING database of differentially secreted regulated proteins, with a known interaction with APP or with the following Gene Ontology terms; neuron projection, synapse, myelin sheath, actin cytoskeleton, IGF regulation (Reactome: mmu 381426), extracellular matrix organization or Alzheimer’s Disease. A. Down-regulated, B. Up-regulated. Abbreviation: FC, fold change. The 

 indicates the protein was classified to be in that category.

(A) Down-Regulated
Symbol	Protein Name	FC	*p*-Value	APP Related	Synapse	Neuron Projection	Myelin Sheath	Alzheimer’s Disease	ECM Organisation	IGF Regulation	Actin Cytoskeleton
CDH13	Cadherin-13	0.003	1.73 × 10^−3^								
GDI2	Rab GDP dissociation inhibitor beta	0.01	2.34 × 10^−2^								
DPYSL3	Dihydropyrimidinase-related protein 3	0.13	4.03 × 10^−2^								
QSOX1	Sulfhydryl oxidase 1	0.22	2.11 × 10^−3^								
CS	Citrate synthase	0.33	8.97 × 10^−3^								
SERPINC1	Antithrombin-III	0.38	6.83 × 10^−4^								
EPHA4	Ephrin type-A receptor 4	0.39	2.36 × 10^−2^								
CKB	Creatine kinase B-type	0.40	3.63 × 10^−2^								
PSMB3	Proteasome subunit beta type-3	0.48	6.94 × 10^−4^								
RPL10A	Large-subunit ribosomal protein l10ae	0.52	1.25 × 10^−2^								
ANXA5	Annexin A5	0.54	3.18 × 10^−2^								
CKM	Creatine kinase M-type	0.55	1.91 × 10^−2^								
CALU	Calumenin	0.56	1.57 × 10^−2^								
ENPP5	Ectonucleotide pyrophosphatase	0.57	2.92 × 10^−2^								
PGK1	Phosphoglycerate kinase 1	0.59	2.11 × 10^−2^								
AGRN	Agrin	0.59	4.81 × 10^−2^								
RPS27A	Ubiquitin-40S ribosomal protein S27a	0.61	3.85 × 10^−2^								
ASRGL1	Isoaspartyl peptidase/L-asparaginase	0.61	4.45 × 10^−2^								
PSMB5	Proteasome subunit beta type-5	0.66	1.06 × 10^−2^								
HSPA4L	Heat shock 70 kDa protein 4L	0.67	4.72 × 10^−3^								
GOT2	Glutamatic-oxaloacetic transaminase 2	0.67	1.38 × 10^−2^								
**(B) Up-regulated**
**Symbol**	**Protein Name**	**FC**	***p*-Value**	**APP Related**	**Synapse**	**Neuron Projection**	**Myelin Sheath**	**Alzheimer’s Disease**	**ECM Organisation**	**IGF Regulation**	**Actin Cytoskeleton**
SPARC	SPARC	1.57	1.88 × 10^−2^								
PDIA3	Protein disulfide-isomerase A3	1.58	2.92 × 10^−2^								
AKR1A1	Aldehyde reductase	1.62	4.35 × 10^−2^								
ITIH2	Inter-alpha-trypsin inhibitor heavy chain	1.65	1.62 × 10^−2^								
CST3	Cystatin-C	1.68	1.20 × 10^−2^								
GNAO1	Guanine nucleotide-binding protein G(o)	1.68	3.98 × 10^−2^								
SPTAN1	Spectrin alpha chain, non-erythrocytic 1	1.72	2.70 × 10^−2^								
CALM3	Calmodulin-1; Calmodulin 3	1.73	1.06 × 10^−2^								
PEBP1	Phosphatidylethanolamine-binding protein	1.73	2.37 × 10^−2^								
DBN1	Drebrin 1; Drebrin	1.73	1.96 × 10^−2^								
PSMB7	Proteasome subunit, beta	1.73	1.40 × 10^−2^								
TPM1	Tropomyosin alpha-1 chain	1.76	3.22 × 10^−2^								
S100A6	Protein S100-A6	1.76	4.37 × 10^−2^								
PAWR	PRKC apoptosis WT1 regulator protein	1.77	1.38 × 10^−2^								
ACTN1	Actinin alpha 1/4 (Alpha-actinin-1)	1.80	2.70 × 10^−2^								
COL5A2	Collagen alpha-2(V) chain	1.85	4.53 × 10^−2^								
HSP90B1	Heat shock protein 90, Endoplasmin	1.85	4.18 × 10^−2^								
RPL14	60S ribosomal protein L14	1.88	1.04 × 10^−2^								
PSMD3	Proteasome 26s subunit, non-atpase, 3	1.89	4.72 × 10^−2^								
COTL1	Coactosin-like 1 (dictyostelium)	1.96	4.92 × 10^−2^								
SCRN1	Secernin-1	1.98	8.80 × 10^−3^								
TPM4	Tropomyosin alpha-4 chain	2.00	3.81 × 10^−2^								
ADD1	Adducin 1 (alpha); Alpha-adducin	2.14	9.37 × 10^−3^								
LDHB	Lactate dehydrogenase B	2.17	1.28 × 10^−2^								
MARCKS	Myristoylated ala-rich C-kinase substrate	2.17	9.25 × 10^−3^								
PSMD6	Proteasome 26s subunit, non-atpase, 6	2.19	2.24 × 10^−2^								
BMP1	Bone morphogenetic protein 1	2.25	3.66 × 10^−2^								
ATP1A1	Na/K-transporting ATPase subunit alpha-1	2.34	2.78 × 10^−2^								
NES	Nestin	2.36	1.04 × 10^−2^								
GALC	Galactosylceramidase	2.49	3.20 × 10^−2^								
SPARCL1	SPARC-like 1	2.65	1.41 × 10^−2^								
FN1	Fibronectin	2.67	2.79 × 10^−2^								
HNRNPA2B	Heterogeneous nuclear ribonucleoprotein	2.67	7.84 × 10^−3^								
PSMB1	Proteasome subunit, beta type 1	2.78	2.27 × 10^−3^								
RPSA	40S ribosomal protein SA	3.10	3.00 × 10^−2^								
HSPD1	60 kDa heat shock protein, mitochondrial	3.39	9.18 × 10^−3^								
NRD1	Nardilysin	4.17	4.66 × 10^−4^								
SERPINH1	Serpin H1	4.31	1.17 × 10^−2^								
TUBB2A	Tubulin beta-2A chain	4.37	2.47 × 10^−5^								
CTSD	Cathepsin D	19.32	9.23 × 10^−4^								
APP	Amyloid-beta A4 protein	23.22	3.41 × 10^−6^								
RPL10	Ribosomal protein L10	23.90	6.53 × 10^−3^								
P4HB	Protein disulfide-isomerase	24.03	1.03 × 10^−2^								
ALDOC	Fructose-bisphosphate aldolase, class i	34.84	3.39 × 10^−2^								
CFL1	Cofilin-1	>100	1.10 × 10^−2^								

**Table 6 ijms-24-07165-t006:** Proteins are regulated in both the astrocyte cellular proteome and the secretome. Seven proteins are found both in the proteome and secretome as differentially regulated, with three * regulated in the same direction and four depleted in the cellular proteome but elevated in the secretome.

Protein	Proteome (Fold Change)	Secretome (Fold Change)
* CKB	0.68	0.40
* DBN1	9.51	1.73
* TPM4	1.59	2.0
COL5a	0.37	1.85
ADD	0.66	2.14
PSMD3	0.59	1.89
HNRMP2b1	0.30	2.67

## Data Availability

The mass spectrometry proteomics data have been deposited to the Proteome Xchange Consortium via the PRIDE partner repository with the data set identifier PXD037902.

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
