# Peer review of "Secreted Amyloid Precursor Protein Alpha (sAPPα Regulates the Cellular Proteome and Secretome of Mouse Primary Astrocytes"

_ijms, 2023, doi:10.3390/ijms24087165_

Round 1
Reviewer 1 Report
“Secreted amyloid precursor protein alpha (sAPPa) regulates the cellular proteome and secretome of mouse primary astrocytes” by Peppercorn et al. provides valuable data on the physiology of sAPPa. They present a massive amount of data supporting the function of sAPPa in astrocytes which is of importance for understanding the pathology of AD in which a shift in the of the non-amyloidogenic pathway to the amyloidogenic pathway is seen. Thus, it is important to know which valuable functions might be reduced when less sAPPa is present.
I have some minor comments and a few major comments which could be taken into consideration:
Minor:
Abstract:
THeoretical = Theoretical
Could it be elaborated on what is meant by “involved with neurologically related functions.”
Introduction:
Line 32: An astrocyte = One astrocyte
Line 57: Could be specified “The content is altered during neurodegenerative disease [23].” = The content of the EVs are altered during neurodegenerative disease.
Line 68: remove (
Line 90: Please elaborate on which genes are being referred to when writing “Some are related to genes previously implicated in Alzheimer’s disease (AD).“ What does this mean? That sAPPa stimulation of astrocytes leads to the upregulation of genes causing AD? Would have expected the opposite.
Results
Line 95: SWATH-MS first time it is mentioned in the text (besides the abstract) please write the name.
Line 204: Twenty-five differentially would write 25 differentially…
Line 219- :” Functional network analysis of the 25 differentially regulated proteins identified from more than one peptide (Supplementary Table 4) revealed a low degree of network 220 connectivity between the proteins with a protein-protein interaction (PPI) enrichment” suggest; “Functional network analysis of the 25 differentially regulated proteins (Supplementary Table 4) revealed a low degree of network connectivity between the proteins with a protein-protein interaction (PPI) enrichment”
Line 269: lack end-bracket (Table 5. = (Table 5).
Discussion
Line 306- “We have identified a group of differentially regulated proteins relevant to neurological function that will provide a focus for further investigation of their relationship to sAPPa” Please specify with examples of proteins relevant to neurological function to make it more tangible.
When writing/finding a link between regulated proteins induced by sAPPa and AD – what kind of link is it? Preventive or causative?
Line 363: What is the perspective of this finding? “It is interesting then, that a group of proteins related to these processes be differentially regulated in astrocytes after sAPPa exposure.”
Methods:
Line 503: Do you have any data on the influence of growing the astrocytes in serum-free cell culture media for 6h? Does it affect cell viability/stress?
For the proteome analysis, were the astrocytes also cultured in serum-free cell culture media for the 2 hours? Please specify in the text.
The method for immunolabelling of the astrocytes is missing, which markers were used?
Figure legends of the supplementary figures are missing.
I think the conclusion makes a nice wrap-up and perspective of the study.
Major:
The hypothesis of the study is mentioned for the first time in the discussion/ conclusion – I would recommend incorporating the hypothesis in the introduction before the aims of the study.
Results:
I think it would be valuable to present the analysis of the cellular composition of the primary mouse astrocyte cultures at the beginning of the result section and include the immunolabeling of the culture in the main article and not as supplementary data. One of the major challenges when working with primary astrocyte cultures is the presence of microglia – have you estimated the amount of these? I guess it is microglia which is labeled in sup fig. 3? Maybe include a few sentences in the discussion about microglia contribution/ or why they are not contributing to the results.
Figure 3: MCL cluster analysis of peptide level proteome data. It is impossible to read the protein names as the text I blurred when zooming in. Consider if the data could be presented in another way showing the data more clearly or if the quality of the figure could be improved.
I think the manuscript would benefit from writing more about the results in the text. Below I have written a suggestion of how the text could be revised “Gene Ontology (GO) categories as well as Reactome and KEGG pathways enrichment analysis revealed categories of importance for neurological functions. This is illustrated in the GO cellular component category 'synapse,' where 48 synaptic proteins were identified;” =
Gene Ontology (GO) categories and Reactome and KEGG pathways enrichment analysis revealed that sAPPa-stimulated astrocytes upregulate the expression of proteins important for neurological function. These proteins are related to the GO cellular component category 'synapse,' where 48 synaptic proteins were identified:”
Please specify the rationale for performing the analysis illustrated in Table 3 and the outcome/results. A sub-selection of the neurologically significant astrocyte proteins regulated by sAPP derived from the proteome peptide level data. Functional pathway analysis using the STRING database highlighted proteins which associated on a (i) first level interaction with APP, (ii) GO terms 190 with neurological function and (iii) Alzheimer’s disease. Abbreviation FC - fold change.
1. All the proteins are expressed by the astrocytes in response to sAPPa and thus how are e.g., 60S ribosomal protein L26 involved in synapse and neuronal projection? The protein 60S ribosomal protein L26 is possibly expressed by all cells having its function intracellularly. Thus, when astrocytes upregulate the expression of 60S ribosomal protein L26 in response to sAPPa stimulation – how would that affect synapse and neuronal projection?
a. This is partly discussed in the discussion. Thus, maybe include a few sentences in the result section about the rationale of the analysis.
2. I think it is understandable/clear-cut with a protein like a superoxide dismutase 1, soluble which the astrocytes can secrete and thus it can influence the function of the neurons.
Reviewer 2 Report
The study by Peppercorn et al. is an interesting study but it is not clear to me what the significance of the study is and how is this study different from the other study by the group. It would be great if authors address my comments when they modify the manuscript for submission to any journal. The authors do not mention the source for the sAPP alpha and do not clearly justify the advantage of using mouse model instead of human model for their study. In addition, it is important that authors check this paper for plagiarism. For ex, the first sentence of the abstract is exactly same as the first sentence of the abstract of https://www.frontiersin.org/articles/10.3389/fnins.2022.858524/full. Overall, I believe these findings could be useful for the field if the significance is clear, for ex, in disease pathology. In the current form, the manuscript is not suitable for publication.
Round 2
Reviewer 2 Report
The manuscript in the current form addresses the concerns raised in previous version and therefore can be accepted for publication.